# Variational Inference with Unnormalized Priors

## Abstract

Energy-based models (EBMs) can be used as powerful priors for Variational Autoencoders (VAEs), improving latent space structure and generative performance. Although previous work has explored EBMs as priors in VAEs, training challenges remain, particularly in efficiently estimating or bypassing the intractable partition function. In this paper, we introduce a generalized training scheme for VAEs with EBM priors and present a comparative analysis of several instantiations of this framework, highlighting their impact on sample quality, likelihood estimation, and optimization stability. By addressing these challenges, we aim to advance the practical applicability of EBM-VAEs and offer insights into their theoretical foundations.

## 1 Introduction

Generative models are essential in unsupervised learning and data generation, with each approach offering unique strengths and facing specific challenges. Among these, Variational Autoencoders (Kingma & Welling, 2014), normalizing flows (Rezende & Mohamed, 2015; Kingma & Dhariwal, 2018), score-based/diffusion models (Song & Ermon, 2019; Sohl-Dickstein et al., 2015; Ho et al., 2020), and energy-based models (Du & Mordatch, 2019; Grathwohl et al., 2020b) represent some of the most influential methods in modern generative modeling. Each of these models brings distinct advantages but also limitations that impact their practical application and effectiveness.

Variational Autoencoders (VAEs) are regarded for their efficiency and scalability. VAEs utilize the variational lower bound (VLB) to approximate complex posterior distributions and have demonstrated considerable success in various tasks such as image generation (Vahdat & Kautz, 2020; Child, 2021) and anomaly detection (Pol et al., 2020). The primary strength of VAEs lies in their ability to efficiently model large datasets through a combination of variational inference and neural network architectures. However, VAEs face a significant challenge due to their use of simple, normalized priors, such as Gaussian distributions. This simplicity can lead to a misalignment between the prior and the posterior, where the model struggles to capture the true complexity and multi-modality of the data. Although efforts to enhance the flexibility of the posterior have been made (Rezende & Mohamed, 2015; Kingma et al., 2016), these methods do not fully resolve other issues pertaining to quality image generation (Dai & Wipf, 2019).

Normalizing flows offer an alternative by applying a series of invertible transformations to a base distribution, allowing for the modeling of complex data distributions with exact likelihood computation. This flexibility makes normalizing flows highly expressive compared to VAEs. However, the challenge lies in designing and training these transformations, which can become computationally demanding and complex, particularly as the dimensionality of the data increases. As a result, while normalizing flows provide powerful modeling capabilities, they may not always be practical for large-scale or real-time applications.

Score-based models and diffusion models (SDMs) represent another innovative approach by learning to model the score function, or the gradient of the log-likelihood, of the data distribution. These models refine noisy data through iterative denoising, leading to high-quality samples and the ability to model intricate data structures. Despite their impressive performance, SDMs face substantial training and sampling challenges.

Training involves optimizing the score function across multiple noise levels, which requires extensive computation. Additionally, the sampling process is typically slow, as generating high-quality samples often involves many iterative refinement steps. These factors can limit the practicality of SDMs for large-scale or real-time generative tasks.

Energy-based models (EBMs) offer a different paradigm by defining probability distributions through an unnormalized energy function. EBMs can capture highly complex and varied data distributions as they parameterize distributions with arbitrary functions, making them extremely flexible compared to other generative approaches. However, the practical application of EBMs is constrained by the need for computationally intensive sampling methods like Markov Chain Monte Carlo (MCMC), which are necessary to approximate the intractable partition function. This reliance on expensive sampling techniques makes EBMs less scalable and efficient compared to other generative models.

Among these approaches, Variational Autoencoders (VAEs) remain our primary focus due to their foundational role in generative modeling and their widespread application in various domains. Being explicit likelihood-based generative models, the main difficulty with VAEs is choosing a parameterization of distributions that are tractable and efficient, yet expressive enough to capture a probability density well. Previous work (Kingma et al., 2016; Grathwohl et al., 2019) attempted this challenge with flexible posteriors, while others explored using powerful decoders (Gulrajani et al., 2017). One particularly interesting approach is to incorporate flexible energy-based priors (Vahdat et al., 2018; Pang et al., 2020; Aneja et al., 2021); EBMs are the most flexible class of generative models in addition to implicit models, which allow them to capture arbitrarily complex latent distributions. However, learning EBMs is notably challenging, which is why it has seen limited uses as priors for VAEs.

In this work, we explore energy-based priors in greater depth. By analyzing energy-based models as priors for variational inference, we derive several different creative approaches, including a generalization, to efficient learning of EBM-VAEs. We explore the various strengths and weaknesses of these approaches, providing greater insight into the use of EBMs as prior distributions for VAEs and highlighting them as potential alternatives to other prior parameterizations.

## 2 Likelihood Estimator for Unnormalized Priors

Let $x \in \mathbb{R}^D$ be some observed data, such as an RGBA image, and $z \in \mathbb{R}^d$ be the associated latent variable. We can represent the relationship through the following joint probability distribution:

$$p_\theta(x, z) = p_\alpha(x|z)p_\beta(z) \tag{1}$$

Where $p_\alpha(x|z)$ is the generative model with parameters $\alpha$ and $p_\beta(z)$ is the prior with parameters $\beta$. We are interested in learning a generative process of the observed data by optimizing $\alpha$ and $\beta$, which can be done by maximizing the log likelihood of the observed data:

$$\ln p_\theta(x) = \ln \int p_\alpha(x|z)p_\beta(z)dz = \ln\left(\mathbb{E}_{p_\beta(z)}[p_\alpha(x|z)]\right) \tag{2}$$

The issue is that evaluating the above expectation directly is generally intractable. One possible solution is to indirectly maximize this by means of variational inference, where we minimize the KL divergence between an approximate posterior distribution $q_\phi(z|x)$, parameterized by $\phi$, and the true posterior distribution $p_\theta(z|x) = \frac{p_\theta(x,z)}{p_\theta(x)}$. This leads to a lower bound on the log marginal likelihood:

$$\ln p_\theta(x) \geq \mathcal{L}_{vae} = \mathop{\mathbb{E}}_{q_\phi(z|x)}[\ln p_\alpha(x|z) + \ln p_\beta(z) - \ln q_\phi(z|x)] \tag{3}$$

We can represent the prior $p_\beta(z)$ in terms of a Boltzmann distribution $\exp(-E_\beta(z))/Z$, where $E_\beta(z)$ is the energy function and $Z = \int \exp(-E_\beta(z))dz$ is the partition function or normalizing constant. The VLB then becomes:

$$\mathcal{L}_{vae} = \underset{q_\phi(z|x)}{\mathbb{E}}[\ln p_\alpha(x|z) - E_\beta(z) - \ln q_\phi(z|x)] - \ln Z \tag{4}$$

The main issue here is the partition function, as it is generally intractable to compute. When training pure energy-based models, samples from the model are required to be generated during training to approximate its gradient. Although high-quality samplers for unnormalized distributions do exist now, they can still have an impact on the training speed and quality of the final model. In the following subsections, we present several potential solutions to make the training of EBM priors more consistent and feasible.

## 2.1 Importance-Weighted Log Partition Function Estimation

To make estimation of the log partition function more practical, we can exploit the approximate posterior to our advantage to obtain self-normalized importance-weighted estimates, leading to the following upper-bounded estimator of the VLB:

$$\mathcal{L}_{vae} = \underset{q_\phi(z|x)}{\mathbb{E}}[\ln p_\alpha(x|z) - E_\beta(z) - \ln q_\phi(z|x)] - \ln\left(\underset{q_\phi(z|x)}{\mathbb{E}}[\exp(-E_\beta(z) - \ln q_\phi(z|x))]\right) \tag{5}$$

After model training, VAEs are often also evaluated using the importance-weighted negative log-likelihood, which is an unbiased estimator of the true negative log likelihood. For finte sample estimates, it gives tighter bounds over the VLB. We may also compute this using self-normalized importance sampling as such:

$$\ln p_\theta(x) = \ln\left(\frac{\mathbb{E}_{q_\phi(z|x)}[\exp(\ln p_\alpha(x|z) - E_\beta(z) - \ln q_\phi(z|x))]}{\mathbb{E}_{q_\phi(z|x)}[\exp(-E_\beta(z) - \ln q_\phi(z|x))]}\right) \tag{6}$$

One can also train energy-based importance-weighted autoencoders (Burda et al., 2016) by directly optimizing Equation (6), which in theory should result in a better generative model.

The gradient of the VLB in Equation (5) with respect to the energy function is as such:

$$\nabla_\beta \mathcal{L}_{vae} = \underset{q_\phi(z|x)}{\mathbb{E}}[-\nabla_\beta E_\beta(z)] + \frac{\mathbb{E}_{q_\phi(z|x)}[\nabla_\beta E_\beta(z)\exp(-E_\beta(z) - \ln q_\phi(z|x))]}{\mathbb{E}_{q_\phi(z|x)}[\exp(-E_\beta(z) - \ln q_\phi(z|x))]} \tag{7}$$

The above gradient resembles the typical log likelihood gradient of an energy based model, with the first term being the positive gradient against the approximate posterior samples. The second term, while not immediately apparent, is in fact the negative gradient under the model, re-expressed as a self-normalized importance-weighted estimate by reusing the true posterior samples. As this is a ratio estimator, the gradient estimator is biased, but consistent; that is, it is asymptotically unbiased in the limit of infinite samples. More importantly, the variance of this estimator decreases with increasing number of samples (Burda et al., 2016). The corresponding standard estimator is as follows:

$$\nabla_\beta \mathcal{L}_{vae} = \underset{q_\phi(z|x)}{\mathbb{E}}[-\nabla_\beta E_\beta(z)] + \underset{p_\beta(z)}{\mathbb{E}}[\nabla_\beta E_\beta(z)] \tag{8}$$

Compared to the traditional gradient estimator, the second term in Equation (5) does not require generating samples from the energy model through MCMC sampling, which can be expensive but also lack convergence guarantees; this makes the importance-weighted gradient estimate appealing, since we reuse the true posterior samples in an asymptotically unbiased estimator.

However, there still exists a major issue with this approach that makes it practically useless. The importance-weighted bound almost always fails because the posterior $q_\phi(z|x)$ does not match the prior but the true posterior, causing it to generate samples that are not representative of $E_\beta(z)$. This inevitably leads to high variance gradient estimates in addition to the ratio-estimate bias in the EBM, causing the prior to converge to a poor solution and density estimates.

## 2.2 Variational Log Partition Function Estimation

The natural solution to this problem is to optimize a second variational lower bound on the normalizing constant $\ln Z$, noting that the proposal distribution $q_\zeta(z)$, parameterized by $\zeta$, explicitly matches the energy model:

$$\ln Z = \ln(\underset{q_\zeta(z)}{\mathbb{E}}[\exp(-E_\beta(z) - \ln q_\zeta(z))]) = \underset{q_\zeta(z)}{\mathbb{E}}[-E_\beta(z) - \ln q_\zeta(z)] + D_{\mathrm{KL}}(q_\zeta(z) \parallel p_\beta(z)) \quad (9)$$

This leads to the following upper bound on the VLB (though it does not result in a bound on $\ln p_\theta(x)$):

$$\mathcal{L}_{vae} \leq \underset{q_\phi(z|x)}{\mathbb{E}}[\ln p_\alpha(x|z) - E_\beta(z) - \ln q_\phi(z|x)] - \underset{q_\zeta(z)}{\mathbb{E}}[-E_\beta(z) - \ln q_\zeta(z)] \quad (10)$$

With this, we end up with an adversarial objective where in the first step, the adversarial prior maximizes the lower bound on the partition function, and then in the second step the EBM prior performs contrastive divergence. Crucially, the variational prior $q_\zeta(z)$ requires a tractable likelihood, which greatly limits the choices of models, requiring either powerful normalizing flows (Kingma et al., 2016; Papamakarios et al., 2017), whose expressiveness is inherently constrained by the computational considerations of computing the Jacobian determinant; or indirect methods that approximate its gradient (Grathwohl et al., 2021). Some methods end up resorting to MCMC sampling (Dieng et al., 2019), which reintroduces the issue we seek to avoid back into the training scheme. This is mainly problematic when modeling in the data manifold, and not so much in the latent dimension as the approximate posterior also pushes itself towards the EBM prior, making the latter's task easier. However, it does still require care as the amortized prior should be powerful enough to be able to keep up with the EBM prior, a property that is not guaranteed.

## 2.3 Generalized Variational Inference Without the Partition Function

Our key takeaway is that neither the inference model $q_\phi(z|x)$ nor the generative model $p_\alpha(x|z)$ depend on the normalizing constant of the prior:

$$D_{\mathrm{KL}}(q_\phi(z|x) \parallel p_\theta(z|x)) = \int q_\phi(z|x) \ln\left(\frac{q_\phi(z|x)}{p_\theta(z|x)}\right) dz$$

$$D_{\mathrm{KL}}(q_\phi(z|x) \parallel p_\theta(z|x)) = \int q_\phi(z|x) \ln\left(\frac{q_\phi(z|x) \int p_\alpha(x|z) p_\beta(z) dz}{p_\alpha(x|z) p_\beta(z)}\right) dz$$

$$D_{\mathrm{KL}}(q_\phi(z|x) \parallel p_\theta(z|x)) = \int q_\phi(z|x) \ln\left(\frac{q_\phi(z|x) \int p_\alpha(x|z) \exp(-E_\beta(z)) dz}{p_\alpha(x|z) \exp(-E_\beta(z))}\right) dz$$

$$D_{\mathrm{KL}}(q_\phi(z|x) \parallel p_\theta(z|x)) = \int q_\phi(z|x) \ln\left(\frac{q_\phi(z|x) \tilde{p}_\theta(x)}{p_\alpha(x|z) \exp(-E_\beta(z))}\right) dz$$

$$D_{\mathrm{KL}}(q_\phi(z|x) \parallel p_\theta(z|x)) = \underset{q_\phi(z|x)}{\mathbb{E}}[\ln q_\phi(z|x) + E_\beta(z) - \ln p_\alpha(x|z)] + \ln \tilde{p}_\theta(x)$$

$$\ln \tilde{p}_\theta(x) - D_{\mathrm{KL}}(q_\phi(z|x) \parallel p_\theta(z|x)) = \underset{q_\phi(z|x)}{\mathbb{E}}[\ln p_\alpha(x|z) - E_\beta(z) - \ln q_\phi(z|x)]$$

The expectation on the right-hand side forms a lower bound on $\ln \tilde{p}_\theta(x)$, which is analogous to the log likelihood of the original VAE objective. However, maximizing this analogous lower bound (ALB) decreases the KL divergence between the approximate and ground-truth posterior distributions, making this a valid variational lower bound for training both $q_\phi(z|x)$ and $p_\alpha(x|z)$ without the dependence on the energy prior's normalizing constant. This independence from the normalizing constant is especially apparent when taking the gradient of the original variational lower bound with respect to the approximate posterior and the generator. This is appealing because the energy prior can be arbitrarily complex and still allow us to efficiently train the approximate posterior and generator.

| **Algorithm 1** Training with Sliced Score Matching | **Algorithm 2** Training with Adversarial CD |
|---|---|
| 1: **repeat** | 1: **repeat** |
| 2: $\quad x \sim D(x)$ | 2: $\quad$ Steps 2 - 4 from Algorithm 1 |
| 3: $\quad z \sim q_\phi(z\|x)$ | 3: $\quad$ Update $\nabla_\zeta \mathbb{E}_{q_\zeta(z)}[-E_\beta(z)] - H(q_\zeta(z))$ |
| 4: $\quad$ Update $\nabla_{\theta\phi} \mathbb{E}_{q_\phi(z\|x)}[\ln p_\alpha(x\|z) - E_\beta(z) - \ln q_\phi(z\|x)]$ | 4: $\quad$ Update $\nabla_\beta \mathbb{E}_{q_\phi(z\|x)}[-E_\beta(z)] + \mathbb{E}_{q_\zeta(z)}[E_\beta(z)]$ |
| 5: $\quad$ Update | 5: **until** converged |
| $\quad \nabla_\beta \mathbb{E}_{q_\phi(z\|x)}[\mathbb{E}_{p(v)}[v^T \nabla_z^2 E_\beta(z)v + \frac{1}{2}(v^T \nabla_z E_\beta(z))^2]]$ | |
| 6: **until** converged | |

However, training the EBM prior does still require the normalizing constant in order to maximize the log likelihood. Here, we note that the EBM approximates the aggregated posterior $q_\phi(z) = \int q_\phi(z|x)p(x)dx$ by reusing latent samples obtained from the VAE posterior. This comes from the fact that one of the terms in an alternative formulation of the VLB (Hoffman & Johnson, 2016; Alemi et al., 2018; Makhzani, 2019) is a KL divergence between the aggregate posterior and prior:

$$\mathcal{L}_{vae} = \mathop{\mathbb{E}}_{q_\phi(x,z)}[\ln p_\alpha(x|z)] - \mathbb{I}_{q_\phi(x,z)}[z;x] - D_{\mathrm{KL}}(q_\phi(z) \parallel p_\beta(z)) \tag{11}$$

With $\mathbb{I}_{q(x,z)}[z;x]$ being the mutual information between the observed data and the latent variable. The negative KL divergence term represents an explicit objective and the only objective of minimizing the discrepancy between the aggregated posterior and the reference prior. Additionally, this KL divergence corresponds to maximum likelihood learning of the energy prior to the aggregate posterior. Here, we propose substituting the KL divergence with other explicit and implicit EBM objectives that similarly match the EBM to a target distribution with or without the need for the partition function, resulting in a much more flexible framework for training energy-prior VAEs.

The key disadvantage of this approach is that there is no guarantee that the prior will be synchronized with the rest of the VAE if alternative learning algorithms are used, so depending on the training algorithm chosen, it may lead to training instabilities.

In this paper, we use sliced score matching, or SSM (Song et al., 2020), to demonstrate matching the EBM to the aggregate posterior, which avoids MCMC sampling by ignoring the intractable normalizing constant. We use SSM because it is simple, stable, and scalable. More importantly, it is a consistent estimator of the target distribution's score, meaning that at optimality, the density represented by the energy prior captures the target density. In contrast, denoising score matching (Vincent, 2011) estimates the score of the corrupted distribution, which would not exactly match the aggregated posterior.

However, we emphasize that any arbitrary objective that pushes the EBM prior toward the aggregated posterior can be used, such as Stein discrepancies (Grathwohl et al., 2020a), as well as standard MCMC sampling and adversarial training (Grathwohl et al., 2021), the latter two correspond to optimizing the original VLB. With this, we now have a generalized framework for training arbitrarily complex EBM VAEs within a rigorously defined theoretical foundation. Algorithm 1 shows the simple process with sliced score matching, and Algorithm 2 demonstrates a similar procedure with adversarial contrastive divergence. We compare and contrast these two instantiations in our experiments.

All of the above training strategies do not make any assumptions about the underlying energy function, so the choice of $E_\beta(z)$ can be arbitrary, ranging from simple restricted Boltzmann machines to large ResNets for higher-dimensional datasets. In this paper, we use Gaussian-Bernoulli RBMs as they are simple but also powerful enough to capture high-dimensional and complex data (Liao et al., 2022). Furthermore, RBMs have already been successfully used as priors for state-of-the-art hierarchical VAEs (Vahdat et al., 2018), and in our preliminary experiments, we found that general neural networks assign infinite energy to latent samples, leading to NaNs at the beginning of training.

The Gaussian-Bernoulli RBM is a specific formulation of restricted Boltzmann machines in which the visible units parameterize a Gaussian distribution, while the hidden units parameterize a Bernoulli distribution, realizing a universal approximator of mixture models (Krause et al., 2013; Gu et al., 2022). The marginal energy of a Gaussian-Bernoulli RBM is as follows (Liao et al., 2022):

$$E_\beta(z) = \frac{1}{2}(\frac{z-\mu}{\sigma})^\top (\frac{z-\mu}{\sigma}) - \text{Softplus}(W^\top \frac{z}{\sigma^2} + b)^\top \mathbf{1} \tag{12}$$

Where $\mu$ and $\sigma$ are the per-visible unit mean and standard deviation vectors, $b$ is the hidden bias vector and $W$ is the weight matrix of the RBM.

To sample from the RBM prior, we can use either the general block Gibbs sampling technique, or take advantage of the continuous nature of the GRBM and utilize Langevin dynamics:

$$z_0 \sim p_0(z), \quad z_{k+1} = z_k - \sigma\nabla_z E_\beta(z_k) + \sqrt{2\sigma}\epsilon_k, \quad k = 1, ..., K \tag{13}$$

Post-training evaluation of the VAE through maximum likelihood estimates is rather challenging, since we cannot entirely avoid the prior's normalizing constant. To that end, there exist approaches to estimating the partition function efficiently. In this paper, we opt for annealed importance sampling (Neal, 1998; Salakhutdinov & Murray, 2008), which is an unbiased estimator of the partition function that is accurate and robust even in high-dimensional spaces. Ratio estimation (Mohamed & Lakshminarayanan, 2017; Mescheder et al., 2017) to estimate the KL divergence between prior and posterior was also considered. However, we found that this produces extremely upper-bounded estimates that are not representative of the true capacity of the model.

## 3 Related Work

Incorporating flexible priors such as energy-based, score/diffusion-based, and mixture priors into variational autoencoders (Vahdat et al., 2021; Han et al., 2020; Lee et al., 2022; Rombach et al., 2022) and also regular autoencoders (Ghosh et al., 2020; Jing et al., 2020) is not a new concept, and has seen some considerable success. Allowing the prior to be learned has many great merits, as it reduces the modeling load on both the posterior and the generative model. A sufficiently powerful prior allows for the use of simpler parameterizations for posterior and generative models, which is appealing in many scenarios.

The idea of matching a learnable prior to the aggregated posterior is motivated by improved maximum likelihood estimation and therefore better generative models (Alemi et al., 2018). A special case of this is the VampPrior (Tomczak & Welling, 2018), which parameterizes a constrained mixture model with the approximate posterior via a finite collection of pseudo-inputs to capture a rough approximation of the true aggregated posterior. Although appealing in theory, VampPriors have their challenges in selecting the best number of mixture components to balance computational efficiency, memory efficiency, and model quality. Energy-based priors do not have such limitations as they can be made arbitrary flexible.

Incorporation of flexible unnormalized priors into the VAE framework is orthogonal to the use of normalizing flows to improve posterior estimates (Rezende & Mohamed, 2015; Grathwohl et al., 2019). Although both strategies aim to make models more expressive, they address considerably different limitations. Normalizing flows focus on improving the posterior distribution $q_\phi(z|x)$ by applying a series of invertible transformations to a simple base distribution (typically Gaussian), introducing more flexibility into the posterior and allowing it to better approximate the true latent distribution. However, normalizing flows are subject to important design constraints in order to ensure computational efficiency, which places a natural limit on how complex or expressive the posterior can be. EBMs do not have such a restriction, allowing for arbitrary flexibility with considerably less restrictions.

Another similar approach is Adversarial Variational Bayes (Mescheder et al., 2017), where the posterior is matched to an arbitrary prior *implicitly* via an adversarial objective. Unlike energy-based priors, which can be trained with any arbitrary objective, the AVB objective is based on black-box density ratio matching,

which requires samples from a predefined prior. Moreover, the variational distribution in AVB is implicit, making posterior inference intractable.

Noise Contrastive Priors (Aneja et al., 2021) are very similar in principle to both AVB and EBM priors in that ratio matching is used to learn a prior. The discriminator in this case represents an exponential tilting of a base distribution, whose optimal value tilts the base distribution to the aggregated posterior. However, while joint training allows the EBM prior to actively shape the VAE, the discriminator in NCP is learned ex post (Ghosh et al., 2020; Dai & Wipf, 2019), i.e., on latent variables of a pre-trained VAE. The latter approach falls short of the key benefits of learning a VAE through an expressive prior, such as improved reconstruction quality and representation learning, as the VAE is otherwise a standard Gaussian VAE.

Most closely related to our work is the generator with a latent-space EBM in Pang et al. (2020). Here, the authors address the exact same problem of learning an energy-based prior through variational inference, making our work orthogonal to theirs. Unlike our work, where we derive a generalized approach to training VAEs with energy priors, the authors focused on a specific instantiation that uses MCMC sampling to learn the EBM. Recognizing that sampling in a pure EBM prior is not guaranteed to converge, they instead resort to modeling the prior as an exponentially tilted distribution, allowing them to train their model reasonably efficiently through short-run MCMC correction of samples from a base distribution. Additionally, albeit for simplicity, the authors also sample directly from the true posterior; we examine amortized inference, which mitigates the costly procedure by jointly learning an approximate posterior, making clever use of the fact that it does not depend on the EBM prior's normalizing constant. The fully amortized model that is superficially discussed, but not empirically experimented with, in their paper is the adversarial instantiation discussed in our framework.

## 4  Experiments

We demonstrate the validity and effectiveness of the proposed training strategies for EBM priors through density estimation on toy and real-world datasets. On toy datasets, we demonstrate the major issue with the importance-weighted partition function estimator and show why it is virtually useless. In all other datasets, we compare and contrast the other training strategies.

### 4.1  Toy Data

In this experiment, we explore the importance-weighted approach to learning EBM priors through several toy datasets. We compare a standard Gaussian VAE with a Gaussian posterior VAE (EVAE) that has learnable variance and a Gaussian posterior VAE with fixed variance ($\sigma = 1$) (ECVAE). For the data, we used 8-Gaussians, checkerboard, and 2-spirals from (Cao et al., 2019) and the four potentials from (Rezende & Mohamed, 2015). We additionally train energy-based VAEs with sliced score matching (SSM) and adversarial contrastive divergence (Adv.), and a VampPrior (Tomczak & Welling, 2018) VAE with 128 pseudoinputs. All models have the same architecture of fully connected DenseNet (Huang et al., 2017) blocks with hidden size of 16 and depth size of 4 in both the encoder and the decoder. Weight normalization (Salimans & Kingma, 2016) is applied to all layers of the encoder and decoder. The Gibbs-Langevin (Liao et al., 2022) sampler is used to generate samples from the EBM VAEs. The results are shown in Table 1.

### Results

Looking at the generated samples and density estimates, it is clear that the unrestricted EVAE failed to capture any of the distributions in any meaningful way. Only by fixing the variance of the approximate posterior to a reasonably large value (resulting in the posterior closely matching the prior) can the ECVAE capture the distributions reasonably well with better samples and density estimates than the vanilla VAE. Note that the arbitrary choice of variance in this artificial example is large enough to allow the ECVAE to explore much of the prior manifold. In contrast, the SSM and adversarial VAEs performed similarly to or better than the regular VAE, especially on the four potentials. The VampPrior VAE performed competitively with the SSM and adversarial VAEs on the 2D data, but failed to capture the potentials, which may indicate possible overfitting of the pseudoinputs.

Table 1: Top: density estimation on toy 2D data (top three are from (Cao et al., 2019)). Bottom: density estimation on four potentials from Table 1 of Rezende & Mohamed (2015)

Again, as noted in (2.1), the posterior does not match the prior, resulting in samples that do not align well with the energy function and leading to poor density estimation. This issue does not exist with the variational partition function estimator, as the variational distribution explicitly favours high entropy and maximizes the likelihood of its samples under the EBM (Grathwohl et al., 2021). To that end, the importance-weighted partition function estimator is virtually useless for any practical scenario.

## 4.2 MNIST

For this experiment, we train fully linear VAEs on dynamically binarized MNIST for 100 iterations. The VAE encoder is composed of sizes 784-256-256-128 with ELUs (Clevert et al., 2016) between, and the decoder is similarly comprised of sizes 64-256-256-784 with ELUs between. The output of the encoder is split into the mean and log-variance of the Gaussian approximate posterior, and the output of the decoder are Bernoulli logits. The prior is a GRBM with 64 visible units and 64 hidden units. The Adam optimizer with learning rates of 0.001 and 0.0002 are used for the VAE and prior respectively. A batch size of 64 for each iteration. We train using four different training schemes: a standard normal VAE; a VAE trained with an adversarial approximate prior, where the adversary is a 4-layer linear inverse autoregressive flow (Kingma et al., 2016); a VAE with a prior trained using sliced score matching; and a VampPrior VAE with 16 pseudoinputs. The samples of the three models are shown in Figure 1, with the EBM VAE samples generated through 100 steps of unadjusted Langevin dynamics. Negative log likelihoods for all models are reported in Table 2. Extended samples for the SSM VAE are available in Appendix A1.

**Results**

Samples from the EBM VAEs are of relatively high quality, indicating that efficient training of VAEs with unnormalized priors in high-dimensional settings is viable. In particular, the successful SSM VAE training validates our generalized EBM VAE training scheme. The variational lower bounds for both EBM VAEs are better than those of the standard VAE, demonstrating that EBM priors can meaningfully improve density estimation. The VLB for the SSM VAE is the best, which may indicate better learning, as the adversarial VAE relies on an additional biased objective on the partition function. The adversarial VAE also took considerably longer to train than the SSM VAE, since an additional network (the adversary) needed to be trained. The VampPrior had slight likelihood improvements over the standard VAE, though it requires a second evaluation of the approximate posterior over pseudoinputs, which is costly and also memory inefficient for large mixture sizes. In general, the SSM VAE demonstrated the best value of the four models.

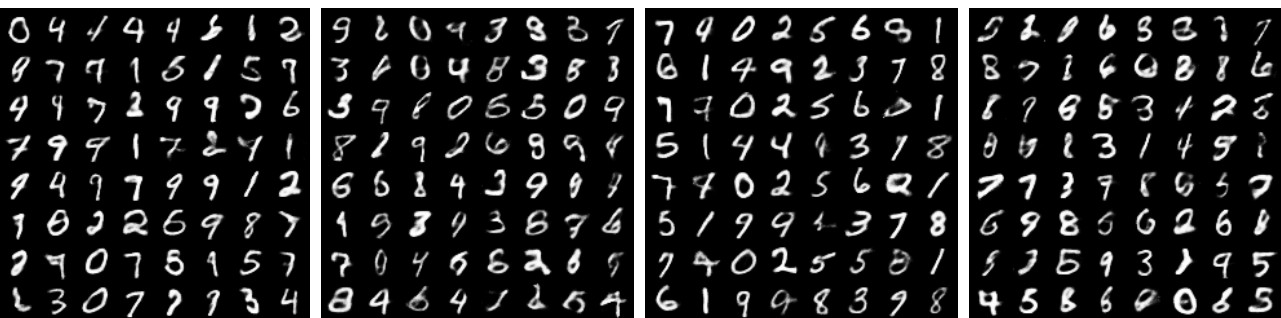

Figure 1: Left to right: Samples from SSM VAE, adversarial VAE, VampPrior and standard VAE.

## 4.3   CIFAR-10

For this experiment, we train a convolutional VAE on 8-bit CIFAR-10 for 100 iterations. The VAE encoder is composed of four 4x4 convolutions of stride 2 and padding 1, with channel sizes of 32, 64, 128 and 256 respectively. ReLU activations are placed after each convolution in the encoder. A linear layer mapping to 512 latent units for both the mean and variance of the approximate Gaussian posterior is the final layer of the encoder. The decoder is composed of four transposed convolutions of channel sizes 128, 64, 32 and 3, all 4x4 kernels with stride of 2 and padding of 1. ReLU activations are placed after all transposed convolutions except the last layer, which maps to the mean of the Laplace distribution. The scale of the distribution is computed analytically using the L1 loss, a technique known as decoder calibration (Rybkin et al., 2021). The prior is a GRBM with 512 visible units and 128 hidden units, which is trained using sliced score matching. The Adam optimizer with learning rates of 0.001 and 0.0002 are used for the VAE and prior respectively. A batch size of 1024 is used for faster training. Similar to the MNIST experiment, we train a standard VAE, two EBM VAEs trained with SSM and adversarial contrastive divergence, and a VampPrior VAE with 16 pseudoinputs. Samples from the prior are shown in Figure 2, and VLB in bits-per-dimension (BPD) are reported in Table 2.

To facilitate calculating BPD, we use the same preprocessing step as (Papamakarios et al., 2017). In short, we first dequantize the images with uniform noise and rescaling to the interval $[0, 1]$ (Theis et al., 2016). Then, we change domains to $(-\inf, \inf)$ by applying a logit transform.

**Results**

Compared to binarized MNIST, the CIFAR-10 dataset is significantly more difficult to solve for generative models due to the massive variation in training samples. All of the VAEs generate reasonable samples but are sub-par compared to the state-of-the-art. This is reflected in the high BPD scores. Interestingly, the standard VAE has lower BPD compared to the two EBM VAEs, suggesting either that the EBM VAEs did not converge and required longer training, or it may indicate learning difficulty. The adversarial VAE had better BPD than the SSM VAE, but it also had blurrier samples, indicating problems with log likelihood as

a measure of generative quality (Theis et al., 2016), but also the fact that score matching does not minimize KL divergence, which may lead to differences in parameters recovered. The VampPrior VAE had the lowest BPD and the highest quality samples, but the sample diversity is relatively low because of the small number of pseudoinputs. Moreover, the VampPrior VAE's BPD score is not significantly better than the normal VAE, which once again indicates that sample quality and likelihood are not mutually inclusive. In general, neither of the EBM learning approaches demonstrated any particular advantage over the standard VAE for CIFAR10, at least when inexpressive architectures are used.

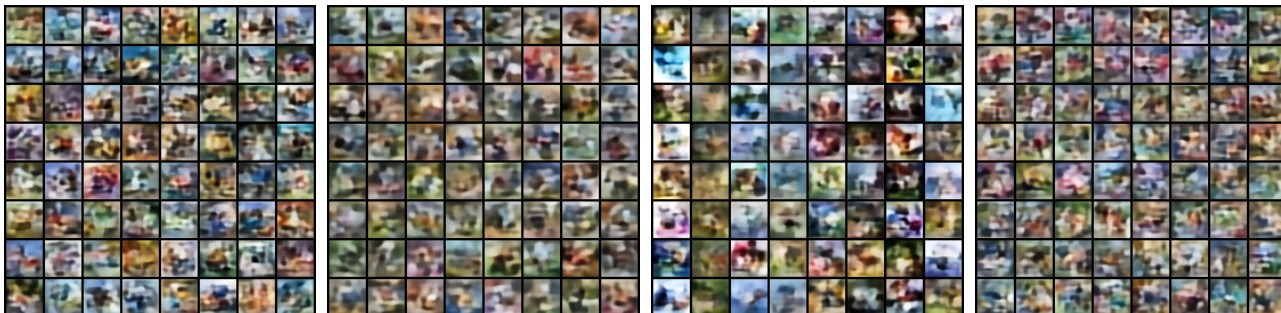

Figure 2: Left to right: Samples from SSM VAE, adversarial VAE, VampPrior and standard VAE.

| Model | MNIST | CIFAR-10 |
|---|---|---|
| NVAE w/o flow (Vahdat & Kautz, 2020) | 78.01 | 2.93 |
| IAF-VAE (Kingma & Dhariwal, 2018) | 79.10 | 3.11 |
| CR-NVAE (Sinha & Dieng, 2021) | **76.93** | **2.51** |
| BFN (Graves et al., 2024) | 77.87 | 2.66 |
| MAF (10) (Papamakarios et al., 2017) | — | 4.31 |
| VAE w/ GRBM prior (SSM) | 91.89 | 6.33 |
| VAE w/ GRBM prior (Adv.) | 92.93 | 6.23 |
| VAE w/ VampPrior | 95.25 | 6.10 |
| VAE w/ $N(0,1)$ prior | 96.03 | 6.13 |

Table 2: Model comparison on binarized MNIST and CIFAR-10 test data. Scores highlighted in bold means best. Top rows are for reference.

## 5 Conclusion

In this paper, we study various approaches to learning VAEs with unnormalized priors. We propose a novel generalization of variational inference that allows flexible training of energy-based priors within a variational autoencoder (VAE) model. We investigated key benefits and limitations of three approaches to learning these priors, bringing new insights into energy-based modeling of VAE priors. We empirically validated these learning algorithms image datasets, showing that energy-based VAEs (EVAEs) learned using alternatives to MCMC contrastive divergence perform well in terms of capturing complex data distributions and producing competitive generative models. Although our experiments primarily focused on Gaussian-Bernoulli RBM priors, the framework is versatile and can be applied to a wide range of unnormalized priors.

## 6 Discussion

Our work on integrating unnormalized priors into the variational inference framework offers a new perspective on generative modeling by bridging the gap between VAEs and energy-based models (EBMs).

Our experiments on the datasets investigate the and nuances in different instantiations of our training framework, particularly in capturing multimodal data distributions compared to standard VAEs. The results

suggest that, provided a good combination of architecture and training scheme, energy-prior VAEs can be very competitive generative models.

Future work could explore alternative strategies for posterior optimization, including hybrid approaches that combine energy-based priors with more expressive posterior distributions, such as normalizing flows. Training unnormalized priors simultaneously with a hierarchical VAE could also potentially improve the representation of complex data structures by conserving energy across layers. Moreover, applying this framework to more diverse types of data, such as audio and text, would provide further insight into the generality and performance of this framework.

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

# A   Appendix

## A.1   Binarized MNIST Linear VAE Extended

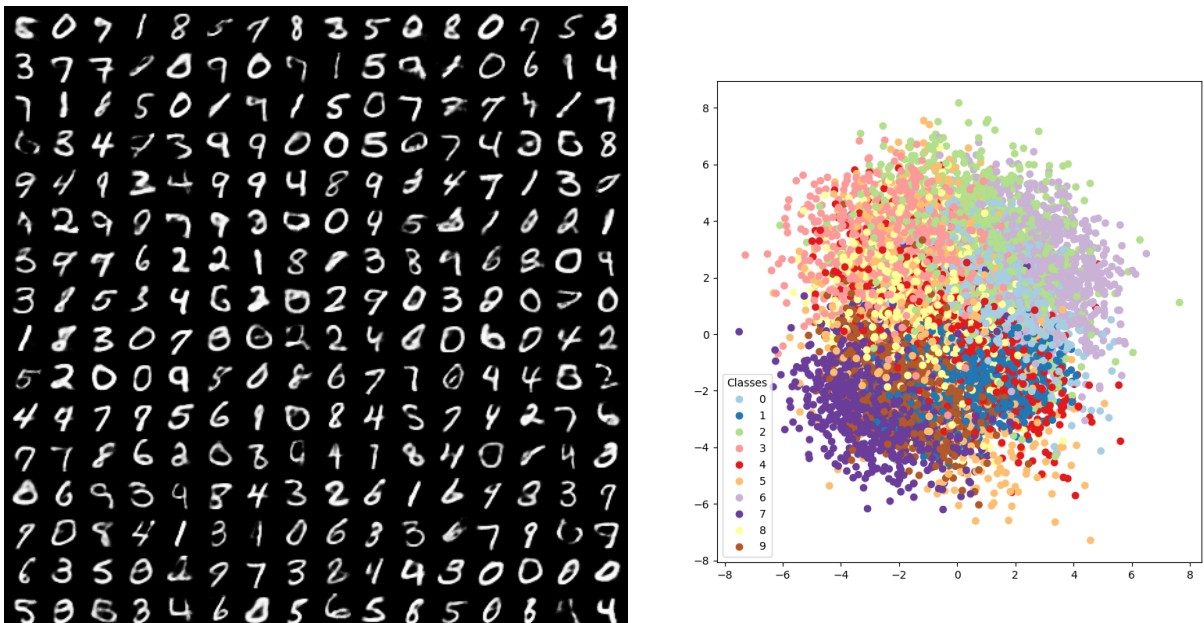

Figure 3: Left: Extended samples from the linear VAE trained using sliced score matching. Right: Latent space (aggregated posterior) induced by the VAE encoder.

