# OpenReview forum: "Variational Inference with Unnormalized Priors"
_TMLR — Rejected by TMLR_

### Review · Reviewer_kUe6 · 2025-03-01

**Summary Of Contributions:**

The paper proposes to adopt an energy-based model as the prior distribution of a VAE, with the goal of improving the flexibility / expressiveness of the model class. It proposes a new training algorithm, where the encoder/decoder parameters are trained with the standard VAE objective and the energy-based prior is trained with a score-matching to match samples from the aggregated posterior distribution. Effectiveness is demonstrated on two small scale image datasets (binarized MNIST and CIFAR).

**Audience:**

Yes

**Broader Impact Concerns:**

None.

**Claims And Evidence:**

No

**Requested Changes:**

- Tighten up the language so it's primarily fact-based and grounded in empirical/theoretical support. See weakness #1.

- Improve clarity of the presentation so it's clear what is being proposed; See weakness #2. Exploring the some of the suggested alternatives and showing how they don't work may also be interesting and add scientific value to the paper.

- Add more comprehensive / larger-scale experiments along with ablations.

**Strengths And Weaknesses:**

Strengths:

The paper explores an yet-unexplored modeling choice of using an energy-based model as the prior in a VAE.

Weakness:

- There's quite some loose and opinion-based statements like *"The core limitation of VAEs lies in their posterior parameterization failing to effectively capture the complexity of the prior distribution"* in the intro section or *"many of these attempts have approached this idea from a fundamentally different perspective hat divorces the objective from theoretically sound probabilistic frameworks (e.g. variational inference), resulting in what is essentially just a 'packaging' of two different models"*, or *"The idea of matching a learnable prior to the aggregated posterior is motivated by the pursuit of safely maximizing mutual information between the latent code and observed data"* in the related works section. IMO they do not belong to a high-quality research paper.

- Lack of clarity in presentation, e.g.:

  - Section 2 of the paper considers a few options for dealing with the unnormalized prior, particularly its log partition function, through variational inequalities/approximations. These options run into various difficulties and are eventually dismissed, before settling on the approach presented in eq (9). I'd prefer if the discussion gets "straight to the point" of the proposed method, or at least make a separation / subsection for the "methods considered but don't work" v.s. the proposed method.

  - Algo box lacks useful details. e.g. Algo box 1 does not mention the training of the prior (e.g., via sliced score matching) at all.

- Experiments feel lacking: despite the supposed generality of the method, it's applied to Gaussian-Bernoulli RBM prior, and only on small scale datasets (binarized MNIST and CIFAR). Some discussions of key design choices are also lacking, e.g., why it's decided to use sliced score matching to train the prior v.s. other methods and how well the method scales.

---

> ### Author Response · Authors · 2025-03-14
> **Edit 1**
>
> Firstly, we would like to thank the reviewer for reviewing our manuscript.
>
> [Major Changes]
> We have made the following major changes:
> - Reevaluated the purpose of our paper; we emphasize that our core contributions are a generalized framework for variational inference with unnormalized priors, and several studies that highlight various instantiations of our framework on various datasets.
> - We discovered a major issue with the way we evaluated log likelihood in the VAEs, which gave us highly optimistic results. We swapped the original approach, which uses density ratio matching to estimate mutual information, with annealed importance sampling to estimate the partition function. The results are now significantly more reasonable and reconciles a lot of the 'inconsistencies' of our framework on larger datasets.
>
> [Weaknesses]
> - The language has been updated to ensure it is fact-based and theoretically/empirically grounded.
> - Presentation clarity is refined, and we compare VAEs trained using sliced score matching and adversarial contrastive divergence, noting that the latter is a special instance of our framework that corresponds to minimizing KL divergence.
> - A standard VAE is trained to offer a fair and balanced comparison.

---

> > ### Comment · Reviewer_kUe6 · 2025-04-05
> > **Thank you for the revision; still room for improvement**
> >
> > I thank the authors for their revised manuscript, which has made significant improvements.
> > The presentation is much clearer, with section 2 now being divided into subsections, and many missing citations added (as also raised by the other reviewers). The revised manuscript has dialed down some of the earlier claims of superiority, instead positioning itself as a study of different methods for training a VAE with an EBM prior; this is reflected in the mixed results, where the EBM prior does not always improve upon a standard Gaussian paper. The scope of experiment is slightly expanded to include toy 2D density estimation problems, and a justification is provided for focusing exclusively on Gaussian-Bernoulli RBM instead of other EBM based prior ("in our preliminary experiments, we found that general neural networks assign infinite energy to latent samples, leading to NaNs at the beginning of training")
> >
> > Still there are some points worth considering/improving:
> > 1. Typo in the 4th equation on page 4 in section 2.3; there's an extra $dz$ appearing in the numerator of the log: $q_\phi(z|x) \tilde p_\theta(x) dz$
> > 2. In the toy experiment (section 4.1), only the importance-weighted approach to learning EBM was studied. I find this unsatisfying, and suggest rerunning the experiment with the other approaches proposed in this paper (as well as closely related ones in the literature). The toy setting is best for demonstrating/understanding the behavior of all the different approaches and comparing their strengths and weaknesses, and not doing so feels like a missed opportunity.
> > 3. The revised paper still left me unconvinced about why one should adopt the EBM prior and further pursue/improve this approach, considering that a simpler approach of learning a parametric prior (say a Gaussian mixture, as in VampPrior) requires no extra variational bounds/approximations, can be trained end-to-end on the ELBO, and might already work better. I echo Reviewer iP7e in suggesting that the VampPrior (or some alternative parametric prior) be included as a reference method in all the experiments.

---

> ### Author Response · Authors · 2025-04-07
> **Revision 2**
>
> We thank the reviewer for responding to our revision.
>
> We have revised the manuscript with the following edits:
> 1. Fixed the typo in the 4th equation in section 2.3
> 2. Expanded the toy experiment to include the other EBM VAEs and VampPrior
> 3. Added VampPrior as a benchmark to binarized MNIST and CIFAR10.
>
> VAE with VampPrior was chosen as the it functions as an approximately optimal solution to the true aggregated posterior. The Related Works section was slightly expanded to discuss the practical limitations of VampPriors, and how EBMs do not face such issues by being arbitrarily flexible.

---

### Review · Reviewer_uspn · 2025-03-07

**Summary Of Contributions:**

The paper proposes a novel method for training variational autoencoders when the prior distribution is parameterized as an energy-based model. Key contributions include:

- A novel variational lower derivation on the log normalizer of the EBM. The derivation starts with the KL between the model and variational posterior. Applying Bayes rule to the model posterior, using the fact that the prior is an EBM, and reorganizing the equation reveals what the paper calls an "analogous lower bound," this one on the log normalizer of the EBM, which is a function only of the variational posterior and generative distribution parameters.

- The paper suggests optimizing the variational posterior q(z|x;\phi) and generative distribution p(x|z; \alpha) parameters independently of the prior EBM parameters by fixing the prior \beta parameters, doing the usual VI maximization, then fitting the EBM parameters to the marginal variational posterior (estimated via Monte Carlo) using an appropriate algorithm of choice.

- Results on MNIST with a judiciously chosen variational and prior families that yields better variational lower bound values than comparable methods.

- Negative results on CIFAR-10 showing worse variational lower bound value than comparable results.

**Audience:**

No

**Broader Impact Concerns:**

None beyond those that apply to all generative models.

**Claims And Evidence:**

No

**Requested Changes:**

The critical weakness indicated above should be addressed before resubmission to this or other venues.

Here are the grounds on which I would have liked to evaluate this proposed method, which is in many ways appealing:
- What are the tradeoffs of using this method compared to other approaches?
- How does this model compare in terms of computational complexity (including all steps, like Langevin sampling) to alternatives?
- What kinds of data will this fit well? Which will it fit poorly? Careful studies of simple distributions should be the first step. This will reveal a lot about where it performs well and where poorly.

**Strengths And Weaknesses:**

Strengths include the bound that yields the optimization objectives, and the results on MNIST.

However, most aspects of the paper do not meet the criteria for acceptance by this journal on the grounds of clarity of narrative and clear presentation arguments.

These critical weaknesses include:
- claims of novelty contradicted by the related works section
- very unclear presentation of the core idea
- profusion of unsupported (empirically or theoretically) claims of superiority of framework
- unclear algorithm, involving a function T that is undefined, and appears to lack dependence on the parameters of interesting, so that the claimed steps do not correspond to the training algorithm as the gradient would be incorrect
- vague and often undefined use of symbols, like \theta and \eta
- a general lack of adherence to standards in mathematical communication that support understanding

Other weaknesses include:
- very early stage empirical validation, which provides mixed evidence at best for the utility of optimizing according to these objectives
- lack of citations to the large body of prior work on such ELBO surgery
- lack of support for the key claimed advantage of this method, that it places the training of EBM priors for VAEs on solid theoretical grounds via variational inference. Note that some of the methods that the paper contrasts itself with do indeed have clear and unambiguous probabilistic justifications in the field of stochastic processes. It is not clear what the authors mean by this.
- poor characterizations of other methods, such as MCMC, which have seen massive advances in recent years
- a mischaracterization of Langevin sampling, used by this paper, which is indeed a variant of MCMC, and goes by the name of the unadjusted Langevin algorithm

---

> ### Author Response · Authors · 2025-03-14
> **Edit 1**
>
> Firstly, we would like to thank the reviewer for reviewing our manuscript.
>
> [Major Changes]
> We have made the following major changes:
> - Reevaluated the purpose of our paper; we emphasize that our core contributions are a generalized framework for variational inference with unnormalized priors, and several studies that highlight various instantiations of our framework on various datasets.
> - We discovered a major issue with the way we evaluated log likelihood in the VAEs, which gave us highly optimistic results. We swapped the original approach, which uses density ratio matching to estimate mutual information, with annealed importance sampling to estimate the partition function. The results are now significantly more reasonable and reconciles a lot of the 'inconsistencies' of our framework on larger datasets.
>
> [Major Weaknesses]
> - Claims of novelty retracted by reevaluating the purpose of the paper.
> - Core idea made more clear, again by reevaluating the purpose of the paper.
> - Claims of superiority redacted, and the language has been updated to ensure it is fact-based and theoretically/empirically grounded.
> - Algorithms are made more clear and explicit.
> - Mathematical symbols are now clearly defined, and an additional introduction of the problem of maximum likelihood learning is discussed.
>
> [Additional Weaknesses]
> - Inconsistencies are now gone after rerunning the experiments with annealed importance sampling.
> - Citations for ELBO surgery are provided.
> - Claims of superiority and foundation are now redacted.
> - Claims made against MCMC methods are now conservative.
> - As far as we are aware, we have not mischaracterized the Langevin algorithm as a non MCMC sampling algorithm
>
> [Requested Changes]
> - We acknowledge that all of the potential solutions to learning EBM priors, including those proposed in this paper, are instantiations of our generalized training framework. We now compare adversarial contrastive divergence, corresponding to minimizing KL divergence, against sliced-score matching (minimizing Fisher divergence) on MNIST and CIFAR10, providing insights into the differences in learning on high dimensional datasets.

---

### Review · Reviewer_iP7e · 2025-03-11

**Summary Of Contributions:**

In this work, the authors propose a new energy-based flexible prior distribution for VAEs in order to improve their reconstruction and density approximation performances. The new approach has two main strengths. First, there is no restriction on the choice of the energy function E. Hence, the resulting prior can be as flexible as desired. Second, the VAE with a such prior distribution can be trained independently from the intractable normalizing constant Z. Finally, the practical interest of this new method is shown on some numerical examples.

**Audience:**

Yes

**Claims And Evidence:**

Yes

**Requested Changes:**

Generally speaking, some points need to be clarified and justify to add strenght to the paper (see Weaknesses section).

Moreover, I would add these two references about the optimal prior distribution (which is $q_\phi$) for Equation (9) and in Section 3:
1. Makhzani, A., Shlens, J., Jaitly, N., Goodfellow, I., and Frey, B. (2015). Adversarial autoencoders. arXiv preprint arXiv :1511.05644
2. Hoffman, M. D. and Johnson, M. J. (2016). ELBO surgery : yet another way to carve up the variational evidence lower bound. InWorkshop in Advances in Approximate Bayesian Inference, NIPS, volume 1.

By the way, I would invert Sections 2 and 3.

**Strengths And Weaknesses:**

**Strengths**

First of all, the main goal of the paper is clear. Moreover, the importance of the choice of the prior distribution in a VAE is well introduced and motivated.

**Weaknessess**

Although the idea sounds interesting, I have a lot of remarks on the paper.

Most of my concerns are located in Section 2. First, from a global perspective, the core of the contribution in Section 2 lacks structure, it is thus not really easy to read. Indeed, I needed several readings to understand the main point of the contribution. In my opinion, the main cause is that the outline of the paper has not been introduced, nor the outline of Section 2. In addition, to add structure to the presentation of the contributions, I suggest to create subsections within Section 2. You should also highlight the main contribution.

Moreover, the understanding of the contribution and of the arguments could be highly improved if each equation is proven. To do so, I suggest you to add an appendix section with the corresponding calculations and proofs.

My main concern is that the justification of the main contribution in page 4 is not convincing enough. Indeed, recall that a VAE aims to maximise the log-likelihood $\log(p_\theta(x))$. Then, by Equations (1) and (2), the analogous lower bound satisfies:

$$\text{ALB}(\alpha,\phi,\beta) = \mathbb{E} [\log p_\alpha(x|z) - E_\beta(z) - \log q_\phi(z|x)] \leq \log p_\theta(x) + \log Z_\beta.$$ However, it is not a lower bound of $\log(p_\theta(x))$. But now you can say that both $q_\phi(z|x)$ and $p_\alpha(x|z)$ are independent from $Z_\beta$. Finally, you can clearly explicit a two-step procedure:
- 1. update $\alpha$ and $\phi$ by maximizing $\text{ALB}(\alpha,\phi,\beta)$
- 2. update $\beta$ by minimizing $D_{\text{KL}}(q_\phi || p_\beta)$

In my opinion, it is a very simpler way to present your contribution. It also answers the question why $\beta$ cannot be updated at the same time than $\alpha$ and $\phi$.  However, since $E_\beta(z)$ is trained in order to make $p_\beta$ close to $q_\phi$ in the sense of the KL divergence, I don't understand why the normalizing constant $Z_\beta$ is not required. Could you explain this point and detail the procedure please?


Then, from a more specific perspective, I have some remarks.
- At the begining of Section 2, I would define the notations and their relations: $\alpha$ for the decoder (or for the likelihood distribution), $\theta$, $\phi$ for the encoder (or the variational posterior distribution) and $\beta$ for the prior. In particular, we have: $$p_\theta(x) = \int p_\alpha(x|z)p_\beta(z)dz.$$
- At the last three lines of page 2, are you talking about Equation (4)? If so, you should precise it. Moreover, Equation (4) is not a lower bound of $\log(p_\theta(x))$ since it is exactly $\log(p_\theta(x))$. You should maybe precise this point.
- To the best of my understanding, the authors of (Burda et al., 2016) do not maximize exactly Equation (4), but a tigher lower bound than VLB. You should maybe precise this point.
- In my opinion, the last paragraph of page 3 is confusing. It adds complexity while reading, even more that the main contribution has not been presented yet. To make it easier to read, I suggest to make it as a remark. Moreover, I think it is worth to precise that a limitation of this approach is that the right-hand side of Equation (8) is not necessarily a lower bound of $\log(p_\theta(x))$ anymore. At last, you should maybe explain a little bit more why this new bound is adversarial.
- The pseudo-codes are not really clear in my opinion. In algorithm 1, I would precise which method is used to train $E_\beta(z)$ and how it is done in practice and cleary explicit the two-step procedure. Moreover, I don't understand the interest of the algorithm 2.


To conclude, I have some final remarks:
- in Section 4, the comparing methods are not defined. It would be nice to do it.
- since the results on the CIFAR dataset are not very good, could you provide some hints to try to improve them?
- I think that a comparison with a VAE using a VampPrior distribution is worth to do, since the ideas of both methods seem close.
- in Section 6, a larger discussion on the limits of the method and/or the difficulty of implementation would be interesting. I would also discuss the influence of the parametrization of $E_\beta$ (multimodal function for example).
- at last, I noticed that almost every references are from arxiv. However, a lot of them have been published in conferences or journals. Therefore, It would be nice to update the references.

---

> ### Author Response · Authors · 2025-03-19
> **Edit 1**
>
> Firstly, we would like to thank the reviewer for reviewing our manuscript.
>
> [Major Changes]
> We have made the following major changes:
> - Reevaluated the purpose of our paper; we emphasize that our core contributions are a generalized framework for variational inference with unnormalized priors, and several studies that highlight various instantiations of our framework on various datasets.
> - We discovered a major issue with the way we evaluated log likelihood in the VAEs, which gave us highly optimistic results. We swapped the original approach, which uses density ratio matching to estimate mutual information, with annealed importance sampling to estimate the partition function. The results are now significantly more reasonable and reconciles a lot of the 'inconsistencies' of our framework on larger datasets.
>
> [Minor Weaknesses]
> - We have defined notations and their relations for our problem
> - Improved clarity in our language to make points more precise
> - Adjusted pseudocodes; we now provide two algorithms for training EBM VAEs with sliced score matching and adversarial contrastive divergence
> - Our results are now significantly more reasonable as a result of changing our evaluation method
>
> For our generalized training scheme for VAEs with unnormalized priors, we argue that our proof does in fact validate the training scheme.
>
> Generally speaking, the VAE maximizes the variational lower bound, which is a lower bound on the log likelihood. However, it is also true that none of the distributions are dependent on each others' normalizing constants. Thus, one can optimize the both approximate posterior and generative model independent of the prior's normalizing constant, which becomes a constant of differentiation. Similarly, the prior is independent of the normalizing constants of the approximate posterior and generative model.
>
> However, as we are also optimizing the prior, the issue of the normalizing constant does persist. Noting that the prior is effectively performing maximum likelihood learning on the approximate posterior's samples to match the aggregated posterior, we can substitute the explicit KL divergence objective with implicit objectives that avoid the normalizing constant altogether. Thus, we have effectively generalized the training of VAEs with unnormalized priors by making it possible to use arbitrary probabilistic objectives to train the EBM prior.

---

### Decision · Action_Editor_TwPY · 2025-04-14

**Recommendation:** Reject

**Comment:**

The paper touches upon an interesting research direction and it should be continued. However, in its current form, it requires more work. The idea of using an energy-based prior in VAEs is not necessarily new, e.g., the reference used in the paper:
- https://arxiv.org/abs/2006.08205

but also other papers that were not commented on or compared to, for instance:
- https://arxiv.org/abs/2209.08739
- https://arxiv.org/abs/2110.10873

I strongly encourage the authors to continue working on the paper and re-submit the paper after a major revision at a later time.

**Audience:**

The paper is of interest to TMLR's audience.

**Claims And Evidence:**

The paper proposes to use an energy-based prior in a VAE. The goal is to improve the expressiveness of the model class. It proposes a new training algorithm, where the encoder/decoder parameters are trained with the standard VAE objective and the energy-based prior is trained with a score-matching to match samples from the aggregated posterior distribution. The model is assessed on two image datasets (binarized MNIST and CIFAR).

The claims of the paper are rather strong:

`By analyzing energy-based models as priors for variational inference, we derive several different creative approaches, including a generalization, to efficient learning of EBM-VAEs.`

As noted out by the reviewers, there is a rather limited evidence to fully support the claims, namely:
- the idea requires more substantial work on formal proofs (the main idea is to train the EBM to minimize its divergence to the optimal prior, i.e., the aggregated posterior, a well-known consequence of the ELBO decomposition [Hoffman & Johnson, 2016; Tomczak & Welling, 2018]);
- it requires careful empirical investigation (empirical results are mixed or even negative, showing EBM prior not consistently outperforming standard (unlearned) Gaussian prior);
- it requires more thorough comparison to related adversarial and deep learning methods.

**Resubmission Of Major Revision:**

The authors may consider submitting a major revision at a later time.